# In Silico Study and Effects of BDMC33 on TNBS-Induced BMP Gene Expressions in Zebrafish Gut Inflammation-Associated Arthritis

**DOI:** 10.3390/molecules27238304

**Published:** 2022-11-28

**Authors:** Farhana Mostofa, Nur Adeela Yasid, Suhaili Shamsi, Siti Aqlima Ahmad, Nur Fatihah Mohd-Yusoff, Faridah Abas, Syahida Ahmad

**Affiliations:** 1Department of Biochemistry, Faculty of Biotechnology and Biomolecular Sciences, Universiti Putra Malaysia, Serdang 43400, Malaysia; 2Department of Cell and Molecular Biology, Faculty of Biotechnology and Bimolecular Sciences, Universiti Putra Malaysia, Serdang 43400, Malaysia; 3Department of Food Science, Faculty of Food Science & Technology, Universiti Putra Malaysia, Serdang 43400, Malaysia

**Keywords:** arthritis, BDMC33, BMPs, in silico study, gut inflammation, TNBS, zebrafish larvae

## Abstract

The bone morphogenic protein (BMP) family is a member of the TGF-beta superfamily and plays a crucial role during the onset of gut inflammation and arthritis diseases. Recent studies have reported a connection with the gut–joint axis; however, the genetic players are still less explored. Meanwhile, BDMC33 is a newly synthesized anti-inflammatory drug candidate. Therefore, in our present study, we analysed the genome-wide features of the BMP family as well as the role of BMP members in gut-associated arthritis in an inflammatory state and the ability of BDMC33 to attenuate this inflammation. Firstly, genome-wide analyses were performed on the BMP family in the zebrafish genome, employing several in silico techniques. Afterwards, the effects of curcumin analogues on *BMP* gene expression in zebrafish larvae induced with TNBS (0.78 mg/mL) were determined using real time-qPCR. A total of 38 identified BMP proteins were revealed to be clustered in five major clades and contain TGF beta and TGF beta pro peptide domains. Furthermore, BDMC33 suppressed the expression of four selected *BMP* genes in the TNBS-induced larvae, where the highest gene suppression was in the *BMP2a* gene (an eight-fold decrement), followed by *BMP7b* (four-fold decrement), *BMP4* (four-fold decrement), and *BMP6* (three-fold decrement). Therefore, this study reveals the role of BMPs in gut-associated arthritis and proves the ability of BDMC33 to act as a potential anti-inflammatory drug for suppressing TNBS-induced *BMP* genes in zebrafish larvae.

## 1. Introduction

The bone morphogenic protein family (BMP) belongs to the transforming growth factor-β superfamily of proteins, which encompasses TGF-beta, activins, growth differentiation factors, inhibins, activin, and nodal and lefty proteins [1,2]. They play a pivotal role [3] during the onset of inflammatory diseases [4]. As a result, several inflammatory disorders, including gut inflammation and arthritis, have pathological features connected to changes in BMP expression and the BMP signalling pathway [5,6].

Gut inflammation is a chronic, recurring disease of the gastrointestinal (GI) tract with an unknown aetiology, which is identified by the inflammation of the mucosal layer and, in some cases, part of the gastrointestinal tract [7]. On the other hand, arthritis is also an inflammatory disease and is connected to chondrocyte hypertrophy and cartilage degeneration [8]. Interestingly, both gut inflammation and arthritis are regulated by pro-inflammatory cytokines. Previous research has shown that an inflammatory gut secretes pro-inflammatory cytokines such as TNF-α that have pleiotropic effects and interfere with several cell-signalling pathways and organ systems [7]. Moreover, IL-17 promotes inflammation in a variety of cell types, including fibroblasts, osteoblasts, monocytes/macrophages, and epithelial cells, as well as the recruitment and activation of neutrophils. In addition, the role of gut microbiota has been shown in arthritis pathogenesis in animal models [9], and it has been observed that gut microbial flora activated immunological factors inside the gut that subsequently triggered joint inflammation. [10]. Altogether, previous studies have mainly focussed on the role of gut microbiota and their metabolites to uncover gut–joint axis relationships rather than addressing the molecular mediators of diseases. 

A plethora of studies have emphasised the significance of BMPs in gastrointestinal disorders [6,11]. During the development of intestinal disease, BMPs play a crucial role as mediators in epithelial–mesenchymal interactions. In addition, a study of BMP gene expression in the colon revealed the presence of *BMP1*, *BMP2*, *BMP5,* and *BMP7* in the top region of the colon crypt [12]. Furthermore, intestinal samples from patients with gut inflammation also exhibited decreased *BMP7* and *BMP5* expression, highlighting the significance of BMPs in this disease [11]. In addition, previous research has revealed that several BMPs, including *BMP2, BMP6, BMP7, BMP9*, and *BMP14,* play significant roles in chondrocyte biology; among them, the levels of *BMP2, BMP6,* and *BMP7* were considerably elevated, which was confirmed by RNA-seq results [13]. In a previous study, [1] *BMP2, BMP4, BMP6* and *GDF5* mRNA expression was reported in healthy and arthritic adult human cartilage. Moreover, *BMP2* and *BMP4* levels were shown to be significantly higher in cartilage lesions in arthritic mice models. [14]. In recent years, genome-wide association studies have made significant strides in identifying the major susceptibility genes and loci [15] for these two diseases [16,17]; however, there is still a lack of knowledge regarding the connection with gut inflammation-associated arthritis.

Trinitrobenzene sulfonic acid (TNBS) is known to induce inflammation in the gut or intestines [17]. It functions as a hapten, binds to tissue proteins to form an antigen, and causes a variety of immunologic reactions. [18]. Previous research has shown that TNBS triggered inflammation in murine and zebrafish models [19]. Consequently, it was assumed that TNBS might also cause inflammation in joints coupled with intestinal inflammation. In addition, TNBS triggers NF-κB and MAPK via an inflammatory cytokine-mediated pathway and initiates signal cross talk with BMPs [20].

Genome-wide studies are now frequently employed to comprehend the involvement of various gene families in the development of inflammatory diseases [21]. According to a genome comparison, over 70% of human genes have at least one zebrafish orthologue [22,23]. Zebrafish embryos were previously assessed as models of chemically induced IBD and arthritis [24]. Thus, using zebrafish models offers an easy assessment of the molecular mechanisms of disease [25]. Additionally, the screening of the gene expression in disease conditions and the evaluation of anti-inflammatory compounds in zebrafish can be performed in a short time and a cost-effective manner [26]. On the other hand, mice lack or overexpress various genes involved in the immune response and rodents have limitations due to time, cost, and ethical constraints. [27]. Meanwhile, in silico and gene expression data obtained from zebrafish in other studies are relevant to human inflammatory diseases [28].

Curcumin is a popular natural compound with several medicinal properties, such as anticancer, antioxidant, antimalarial, and anti-inflammatory properties [29]; however, it suffers from certain poor pharmacological properties, for example, its poor bioavailability and bio-absorbability due to presence of a β-diketone moiety and an active methylene group [30]. Their presence makes curcumin’s structure unstable [31] and allows curcumin to be quickly destroyed by aldo-keto reductase in the liver at around 176–177 °C [32]. To overcome this shortcoming, our research group synthesized curcumin derivatives, namely, BDMC33, by eliminating these groups. The newly synthesized curcumin analogue contains three aromatic rings instead of the two aromatic rings in curcumin [33] and has also undergone other structural modifications, notably, the addition of halogen groups, which enhanced its pharmacological properties [34]. In addition, the Papp values for curcumin were 0.10–6 cm/s, whereas for BDMC33 they were increased up to 2.0 × 10^–6^ cm/s, demonstrating that BDMC33 is a highly absorbable molecule [35]. Moreover, BDMC33 showed promising anti-inflammatory activities, which are primarily mediated by modulating a wide range of transcription factors, cytokines, protein kinases, adhesion molecules, redox states, and enzymes known to be involved in the inflammation process [36].

Therefore, in this study, we undertook a genome-wide study of the BMP family in zebrafish using several in silico techniques. We hypothesized that BMPs might be a common player during the onset of gut inflammation-associated arthritis. We also determined the expression of selected BMP genes using the qPCR technique and finally evaluated the anti-inflammatory activity of BDMC33 towards TNBS-induced zebrafish larvae in modeling gut inflammation-associated arthritis.

## 2. Results

### 2.1. Genome-Wide Analysis of BMP Family in Zebrafish

In this study, a total of 38 BMP proteins were identified containing either TGF beta or TGF beta pro peptide domains in the zebrafish genome. A domain analysis showed that this family’s proteins have a similar domain organisation, in which the predicted TGF motifs were found to be located near the C-terminus of each protein (Figure 1). Among these proteins, four proteins (Lefty 1, Lefty 2, GDF8, and Myostatin) contained only four motifs. Furthermore, BMP2b and GDF6-A-like proteins had two motifs predicted. In addition, five consensus motif sequences were predicted in this analysis. However, three motifs (Motif I: C-X-G-X-C, Motif II: C-X8-F, and Motif V: C-X-C) showed high conservation of cysteine, glycine, and phenylalanine residues at certain positions in the domains of all BMP proteins (Figure 2).

### 2.2. Gene Structure Analysis of BMP Family in Zebrafish Genome

The coding regions of these zebrafish BMP genes were encoded for amino acid (aa) sequences with a length of 347–501 residues in which the shortest protein was BMP7b (347aa), and the longest protein was BMP6 (501aa) (Figure 3). Furthermore, a phylogenetic relationship analysis inferred that these 38 proteins were clustered into five major clades (I–V), as shown in Figure 4. Each major clade contained five proteins, except clade IV with six proteins. Clade I contained GDF5, GDF6A, and GDF10 as well as BMP3 and ADM proteins. This tree also grouped BMP5, BMP6, BMP7a, BMP7b, and BMP8 in clade II. In clade III, Nodal-related 2, Nodal-related 1, GDF6 A-like, BMP15, and GDF9 proteins are clustered together. Meanwhile, clades IV and V consist of entirely TGF and inhibin proteins that are clustered distinctly. However, Southpaw protein was clustered together with TGF proteins in clade IV (Figure 4).

### 2.3. Physicochemical Properties of BMP Family of Zebrafish Genome

From the physicochemical results’ analysis, it was found that the molecular weight of the 38 proteins ranges from 40,201.62–54,217.72 Da. Based on the computed PI values, most of the proteins are basic in nature and contain more positively charged residues (1883) than negatively charged residues (1605). According to instability index (II) data, most of the proteins have values more than 40, meaning that they are unstable [30]. In line with the II values, the Aliphatic index values of the proteins are high (71.07–90.28), indicating these BMP proteins are thermally stable. The grand average of hydropathy (GRAVY) score of the proteins represents the hydrophobic nature of the proteins with good solubility and it was found that all proteins have negative values (−0.225−(−0.786)) for their GRAVY scores, so they are hydrophilic in nature and possess isoelectric point (pI) values between 6.13 and 9.85. These pI values showed that 83% of the proteins are basic in nature (Table 1). Finally, we used these genome-wide-analysed data to choose prominent BMPs related to gut inflammation-associated arthritis.

### 2.4. Optimization of TNBS Concentration

The survival rate of the zebrafish larvae treated with TNBS at multiple concentrations (0.2–50 mg/mL) was observed at 9–12 dpf. The zebrafish larvae at 12 dpf showed a high survival rate (>60%) at a concentration <0.78 mg/mL (Figure 5). Meanwhile, the untreated larvae showed a 100% zebrafish larvae survival rate. Based on the mortality rate, the LC_50_ value of TNBS towards the zebrafish larvae at 12 dpf was 0.60 mg/mL (Figure 6). However, for our gene expression analysis, we used TNBS at a concentration of 0.78 mg/mL to induce inflammation in the larvae and evaluate the anti-inflammatory effect of BDMC33 on selected BMP genes in zebrafish larvae at 9–12 dpf, since the survival rate was >60%.

### 2.5. Relative Normalized Fold Expression of BMP7b, BMP4, BMP2a, and BMP6

In this study, the zebrafish larvae were exposed to TNBS at a concentration of 0.78 mg/mL for 3 days (9 dpf to 12 dpf) to induce inflammation and measure the gene expression of *BMP2a, BMP4, BMP6,* and *BMP7b*. Among the four genes, *BMP2a* showed the highest-fold increment (17-fold), which was followed by *BMP7b* (6-fold), *BMP4* (4-fold), and *BMP6* (2-fold) (Figure 7). All gene expressions were normalized against two reference genes: *GAPDH* and *β-actin*.

### 2.6. BDMC33 Suppressed Gene Expression of BMP2a, BMP4, BMP6, and BMP7b in Inflammatory State

To evaluate the effects of BDMC33 on the TNBS-induced larvae, we determined the expression patterns of the *BMP2a*, *BMP4*, *BMP6, and BMP7b* genes. In this study, the expression of the four *BMP* genes was downregulated in the TNBS-induced larvae with the presence of BDMC33. The *BMP2a* gene (eight-fold) showed the highest degree of suppression at a concentration of 1.25 µM (Figure 8a), while *BMP7b* showed the highest suppression at a concentration of 0.25 µM (four-fold) (Figure 8b), *BMP4* showed at the highest concentration of 6.25 µM (four-fold) (Figure 8c), and *BMP6* (Figure 8d) was inhibited at 1.25 µM (three-fold) compared to the control group. Interestingly, the downregulation of *BMP7b* showed a dose-dependent concentration pattern; however, the other three genes showed downregulation at different concentrations.

## 3. Discussion

It has been well-established that several diseases are coordinated by the active participation of BMP family members [37]. Thus, in this study, we performed a genome-wide analysis of the BMP family using the zebrafish genome. Through our in silico analysis, 38 bone morphogenic proteins were identified from a phylogenetic tree analysis. The results of the domain and motif and gene structure analyses of the BMP family showed that zebrafish and humans share the same conserved domain: either the TGF-β peptide or the TGF-β pro peptide. The sequence logo revealed that the signature of the motif was enriched with the cysteine, glycine, and phenylalanine amino acids, which is similar to the human BMP protein motif [16]. Human BMP proteins are glycoproteins of relatively low molecular mass whose calculated isoelectric point (pI) ranges from 8–10 [7]; these values are similar to our findings. Therefore, the BMP family of both humans and zebrafish are closely related with respect to their genetic features.

TNBS is a known inducer of the intestines that modulates the secretion of proinflammatory cytokines and triggers inflammatory responses. Previous studies have shown that TNBS increases intestinal BMP expression in inflamed larvae [10,20]. In our study, the expression levels of the *BMP2, BMP4, BMP6,* and *BMP7b* genes were increased significantly due to acute inflammatory reactions triggered by TNBS. It was believed that the upregulation of BMPs simultaneously initiated an inflammatory cascade in both the gut as well as the cartilage and bone joints. In our study, the expression of the *BMP2* gene was upregulated, which was consistent with a study conducted in a murine colitis model where TNBS was used as an inducer [11]. Furthermore, higher *BMP2* expression in TNBS-induced colonic inflammation might be coupled with decreased sensitivity or lack of inhibitory factors [6,38] leading to enhanced epithelial cell proliferation [7]. In addition, the upregulation of *BMP7b* expression activates the *MMP13* gene [39] for extracellular cell degradation in cartilage and bone joints, which causes cartilage degeneration [40]. Moreover, *BMP2, BMP4,* and *BMP6* were also upregulated, which might interfere with interleukin IL-1β production and cause subchondral bone changes [41].

A prior work on zebrafish utilising TNBS-induced mice found increased expression of *BMP7* in the acute phase and *BMP6* in the chronic phase of colitis [42]. *BMP4* expression was similar in the early and late stages of colitis, whereas *BMP2* expression was considerably higher in the acute stage [6]. In addition, *BMP2* and *BMP6* expression were also upregulated in chondrocytes hypertrophy in another study [13]. The expression level of the *BMP7* gene as reported in [43] knee osteoarthritis (OA) patients was substantially higher than in healthy controls. Moreover, in a mouse model of OA, [44] the detection of a strong upregulation of *BMP2* and *BMP4* in neighbouring cartilage lesions was reported [45]. However, there is little available knowledge on how TNBS interacts with the BMP family and induces inflammation in the gut. In a previous work [11], it was reported that there was an increased expression of *BR-Smad,* including *Smad3* and *BMP4,* during the exposure of TNBS to murine models. Hence, to provide a possible explanation from our study, it is predicted that TNBS binds to the TLR4 receptor of the NF-κB pathway and transduces the signal through a MyD88-dependent manner [46] and initiates cross talk with the BMP and MAPK pathways; thus, the BMP pathway undergoes multiple phosphorylation events and activates the R-Smad (*Smad1/5/8*) complex [47,48]. Meanwhile, phosphorylated *Smad1/5/8* activates co-Smad (*Smad4)* and forms a complex that translocates to the nucleus [47] where the complex is further coupled with coactivators of the MAPK and NF-κB pathway to regulate downstream gene expression and increase the expression of intestinal BMPs by activating an inflammatory cascade, which leads to the upregulation of skeletal BMPs as well.

Curcumin derivatives have recently showed promising anti-inflammatory properties by blocking a number of key inflammatory mediators, including tumour necrosis factor-α (TNF-α), interleukin (IL)-6, interleukin (IL)-1β, nitric oxide (NO), and nitric oxide synthase (iNOS) [49]. The chemical structure of curcumin, which includes several substituents such as methoxy, hydroxyl, alkyl, halogens, amino, nitro, nitril, carboxyl, and benzene groups in its structure, is largely responsible for its therapeutic potentials [50]. Recent research has demonstrated that the structural alteration of curcumin increased its anti-inflammatory efficacy due to its increased stability in aqueous solutions [30,39]. Lee et al. [47] has reported that BDMC33 successfully reduced the production of NO, TNF-α, and IL-1β in LPS-induced macrophages. Additionally, the inhibitory effects of BDMC33 were associated with the suppression of AP-1 DNA-binding activity by halting the JNK and ERK signalling pathways, as well as the inactivation of NF-ĸβ signalling, which occurred due to the attenuation of I-ĸB degradation, I-ĸB phosphorylation, NF-ĸβ translocation, and DNA binding [48]. In association with the present study, it is assumed that BDMC33 interacts with a noncanonical BMP signalling pathway that initiates upon the binding of TNBS to the TLR4 receptor and transduces a signal in association with MyD88, thereby activating IKK MAPK [51] and initiating cross talk with canonical members (Smad 1/5/8) of BMP signalling to inhibit downstream phosphorylation [15] and the nuclear translocation of I-ĸB, ERK/JNK, and the Smad1/5/8-Smad4 complex [52] and attenuate the upregulation of the *BMP2a, BMP4, BMP6* and *BMP7b* genes. Overall, both the canonical and noncanonical pathways of BMP signalling are concerted in this process in which a protein kinase, MAPK, and a transcription factor, NF-κB, play a key role to suppress the inflammatory state of the *BMP2a, BMP4, BMP6,* and *BMP7b* genes. To our knowledge, this is the first time BDMC33 was used to suppress BMP genes in an inflammatory state. Thus, future studies are needed to address how BDMC33 interacts with the BMP family and attenuates inflammation. In addition, it is important to note that from our in-silico analysis, we have identified 38 BMP family members; however, we only focused on four genes, namely, *BMP2a, BMP4, BMP6,* and *BMP7b,* in a gut inflammation-associated arthritis gene expression study.

Taken together, a further investigation and data analysis need to be conducted to validate the effect of BDMC33 on a gut inflammation-associated arthritis model targeting the BMP family; nonetheless, this preliminary study of the effect of BDMC33 in gut inflammation-associated arthritis developed in zebrafish and targeting the BMP family could be used to establish a primary genetic correlation of the gut–joint axis.

## 4. Materials and Methods

### 4.1. Animal Care and Husbandry

The Universiti Putra Malaysia’s Institutional Animal Care and Use Committee (IACUC) has approved this in vivo study (UPM/IACUC/AUP-R044/2022), which was conducted in compliance with said committee’s regulations. Several adult AB strain zebrafish (*Danio rerio*) were brought from Danio Assay Laboratories Sdn. Bhd. (Malaysia) and grown in the Animal Biochemistry and Biotechnology lab of the Faculty of Biotechnology and Bimolecular Sciences, UPM. In order to stimulate spawning, mature male and female zebrafish were allowed to acclimate to a 14 h light/10 h dark photoperiod cycle at 28 °C in a dechlorinated circulating aquarium system for a week. Zebrafish adults were given a morning meal of flakes and an evening meal of brine shrimp (*Artemia salina*) every day. Fertilized eggs were collected 1 h after light was turned on from a collection container that had been set up near fake aquatic plants to mimic spawning areas. Before being used in any experiment, embryos were washed three times with embryo medium after being washed once with distilled water.

### 4.2. Identification of BMP Family

Human BMP7 protein sequence (Accession: NP_001710.1) was downloaded from the NCBI database [53]. This protein sequence was used as reference to identify Hidden Markov Model (HMM) profile of BMP family in the Protein family database (pfam) [54] in which TGF beta pro peptide and TGF beta family were shown to be the highest matched domains. These HMM profiles were used in HMMER database [55] to identify all BMP family proteins in the zebrafish genome. Later, Conserved Domain Database [56] and BLASTP program [57] were used to validate the identified protein sequences of BMP family.

### 4.3. Sequence Analyses of Zebrafish BMP Family Members

The MEME (Multiple EM for Motif Elicitation, v.4.9.0) software developed by Timothy Bailey (University of Nedeva, Virginia, NV, USA) and William Stafford Noble (University of Washington, Seattle, WA, USA) was used at first to determine the conserved motifs and sequence logo of protein domains in the BMP family of zebrafish. Then, the genomic and coding sequences of zebrafish BMP genes were obtained from the NCBI data hub for gene structure analysis. Next, BMP gene structure was predicted using Gene Structure Display Server 2.0 (http://gsds.cbi.pku.edu.cn/ (accessed on 15 March 2022)) developed by Center for Bioinformatics (CBI), Peking University, China web tool. Furthermore, all BMP protein sequences were aligned in MEGA (Molecular Evolutionary Genetics Analysis, v.6.0) developed by Panne state University, USA using the MUSCLE algorithm with the default parameters [58]. The maximum likelihood approach with 1000 bootstrap replicates was then used to generate an unrooted phylogenetic tree. Finally, the ExPASy ProtParam programme (https://web.expasy.org/protparam/ (accessed on 15 March 2022)) developed by Swiss Institute of Bioinformatics, Switzerland was employed to compute the physicochemical parameters of BMP proteins. 

### 4.4. Optimization of TNBS Exposure to Larvae

Five percent (*w*/*v*) of 2,4,6-trinitrobenzene sulfonic acid was purchased from Sigma Aldrich (Cat# P2297). First, TNBS was diluted in E3 media with 2-fold serial dilutions in a 96-well plate. Then, larvae aged at 9 dpf were exposed to TNBS at different concentrations ranging 0.20–50 mg/mL to determine survival rate at 12 dpf. Finally, LC_50_ value was calculated from mortality rate at 12 dpf. Untreated larvae were used as controls.

### 4.5. BDMC33 Preparation 

Curcumin derivative, BDMC33 or 2,6-bis(2,5-dimethoxybenzylidene) cyclohexanone, was chemically synthesized at the Institute of Bioscience, Universiti Putra Malaysia [34]. Stock compound was prepared at the concentration of 50,000 µM in 100% DMSO. Working solutions were then prepared in five-fold serial dilutions at 6.25, 1.25, and 0.25 µM in 0.2% (*v*/*v*) DMSO.

### 4.6. Total RNA Extraction and cDNA Synthesis

Thirty 12-day-old larvae were pooled from each of uninduced, TNBS-induced, and TNBS-induced with treatment (BDMC33) groups for total RNA extraction using Monarch^®^ Nucleic Acid Extraction kit (New England Biolabs, UK) according to the manufacturer’s instructions. RNA quantity and quality were measured using a Nanoquant infinite M200pro (Grödig, Austria) and 1% (*w*/*v*) agarose gel electrophoresis, respectively. Reverse transcription was performed to synthesize cDNA from 0.5 µg of extracted total RNA using the Viva cDNA Synthesis kit (Vivantis, Malaysia) according to the manufacturer’s instructions.

### 4.7. Gene Expression Analysis

In this study, four BMP genes (BMP2a, BMP4, BMP6, and BMP7b) were selected to analyse the expression patterns of BMP family in the presence of curcumin analogue. Specific primer pairs of the target BMP genes and reference genes were designed using NCBI database [59] and validated using IDT Oligoanalyzer [60]. All primers (Table 2) were purchased from IDT, Malaysia. Quantitative real-time polymerase chain reaction (qRT-PCR) was performed using ViprimePLUS Taq qPCR Green Master Mix (SYBR^®^Green Dye) (Vivantis, Malaysia) according to the manufacturer’s instructions. The qRT-PCR program was set as follows: 95 °C for 2 min and 40 cycles of 95 °C for 15 s, specific annealing temperature applied for 30 s (Table 2), and final extension at 72 °C for 30 s in BIORAD CFX96™ Real-Time system (Hercules, CA, USA). Every experimental run included a routine melt curve analysis to identify any non-specific PCR products. The melt curve analysis data were collected during the increase in temperature from 65 °C to 95 °C with a ramp rate of 0.5 °C/s. Relative normalised expression was calculated using Livak method [61], where *GAPDH* and *β-actin* were used as reference genes in the BIORAD CFX-96 manager 3.1 software. In this calculation, uninduced samples were used as calibrators to normalize induced and uninduced samples, while induced samples were used as calibrators to normalize uninduced, induced, and treated samples in respective calculations. 

### 4.8. Statistical Data Analysis 

Except where otherwise specified, all results were reported as the mean S.E.M. (Standard error of the mean) of three replicates. One-way analysis of variance (ANOVA) followed by Dunnett test were used to determine statistically significant differences among the groups. Statistically significant differences from the control group were found at the * *p* < 0.05, ** *p* < 0.01, and *** *p* < 0.001 levels. All graphical representations used in this paper were constructed using GraphPad Prism, version 8.1, (GraphPad Software, Inc. San Diego, CA, USA).

## 5. Conclusions

In summary, we have identified 38 BMP members from zebrafish containing either TGF-β peptides or TGF-β pro peptides, which are similar to human BMP family members. In this preliminary study, we have shown that *BMP2a*, *BMP4*, *BMP6,* and *BMP7b* gene expression was upregulated in the TNBS-induced larvae, which indicates a probable relationship with gut inflammation-associated arthritis as those four genes play a pivotal role during the onset of the two constituent diseases. Furthermore, the curcumin analogue BDMC33 downregulated all four BMP genes in the TNBS-induced zebrafish larvae and exhibited its anti-inflammatory activities. Therefore, these findings might provide a starting line for understanding the relationship constituting gut inflammation-associated arthritis targeting the BMP family as well as the potential of BDMC33 as a new drug candidate in treating inflammatory diseases.

## Figures and Tables

**Figure 1 molecules-27-08304-f001:**
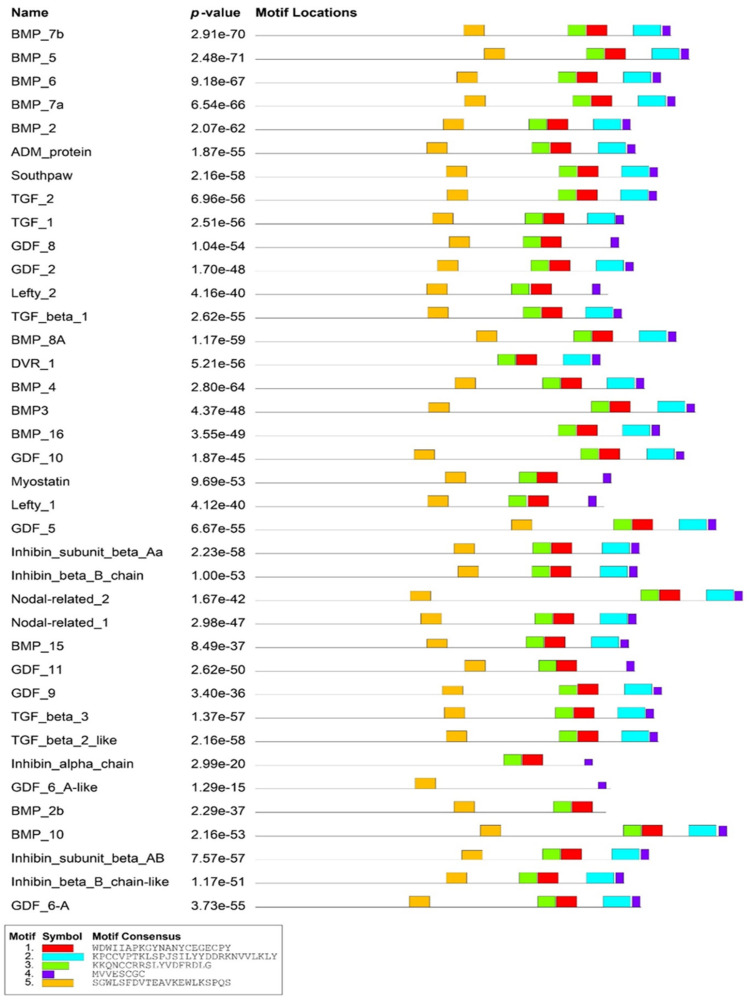
Domain organisation of 38 BMP family members. Five consensus sequences of TGF motifs were identified in these proteins. Red, cyan, and green boxes indicate TGF beta peptide domains. Yellow box indicates TGF beta pro peptide domain. TGF = Transforming growth factor.

**Figure 2 molecules-27-08304-f002:**
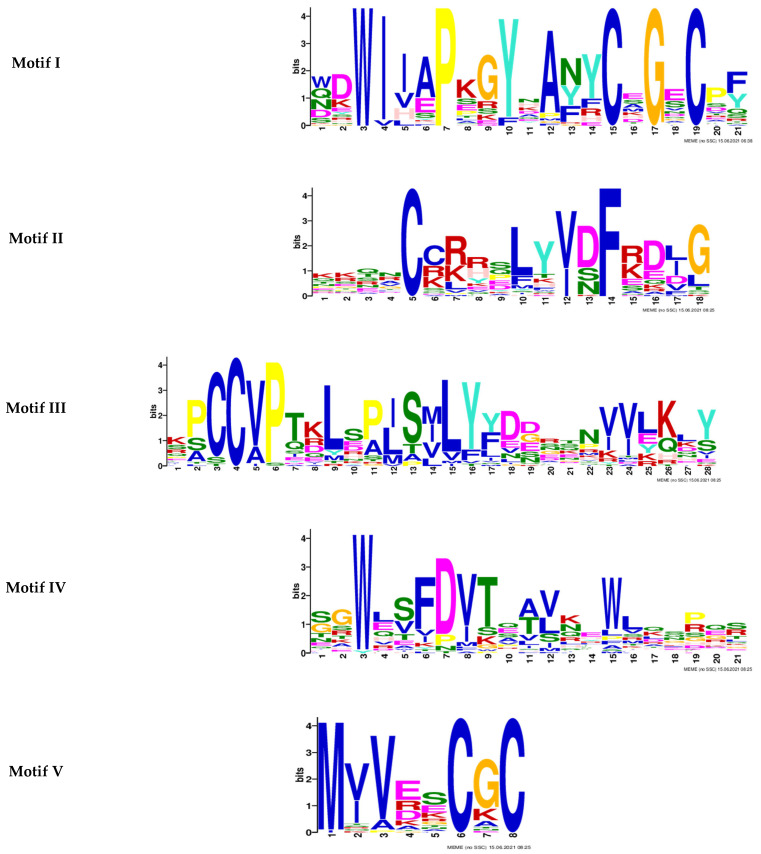
Sequence logo of five motifs predicted from the 38 BMP family proteins of zebrafish. Cysteine, glycine, and phenylalanine residues were identified to be prominent in the motifs.

**Figure 3 molecules-27-08304-f003:**
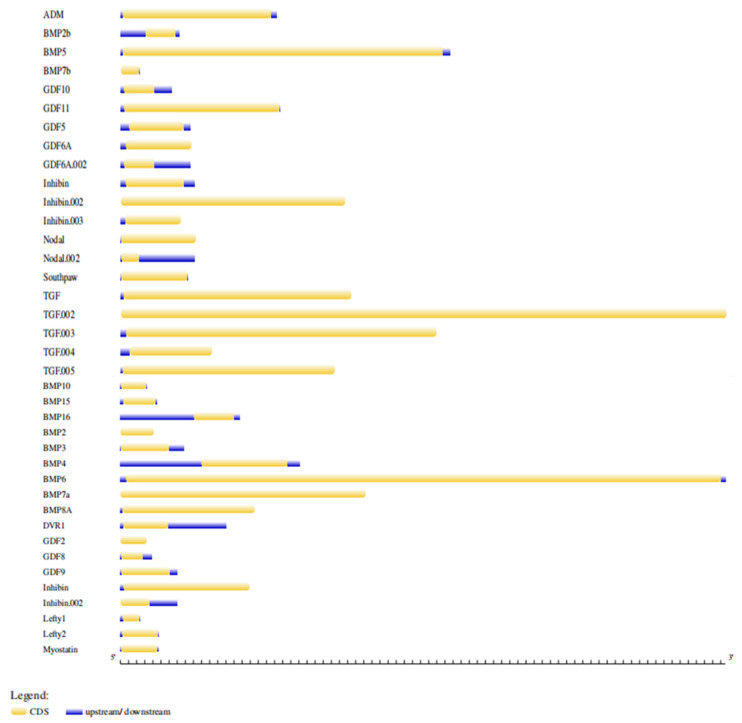
Gene structure of 38 zebrafish *BMP* sequences. ADM = Adrenomedullin, BMP = Bone morphogenic protein, GDF = Growth differentiation factor, TGF = Transforming growth factor, DVR1 = decapentaplegic-Vg-related 1, and Lefty = Left-right determination factor.

**Figure 4 molecules-27-08304-f004:**
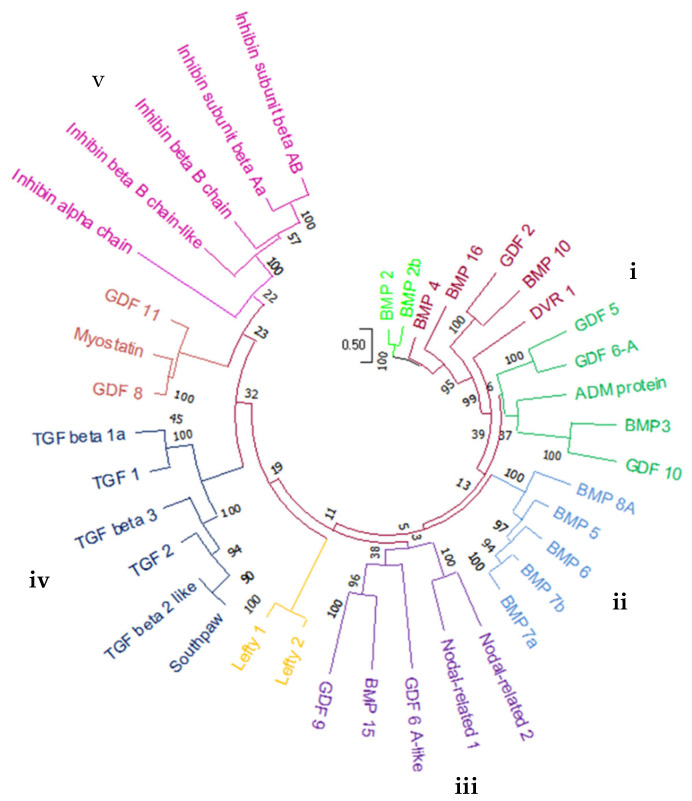
Phylogenetic relationship of 38 BMP members identified in zebrafish genome. Phylogenetic tree was produced using a maximum likelihood (ML) method with 1000 bootstrap replicates in MEGA-X software. I–V represent five major clades.

**Figure 5 molecules-27-08304-f005:**
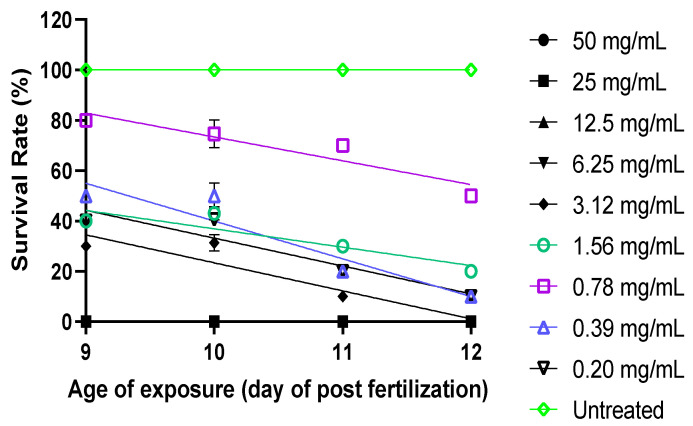
Survival rate of zebrafish larvae (9 to 12 dpf) exposed to TNBS at two-fold serial dilution (0.2–50 mg/mL). Zebrafish larvae at 12 dpf survived (>60%) in TNBS at concentration <0.78 mg/mL.

**Figure 6 molecules-27-08304-f006:**
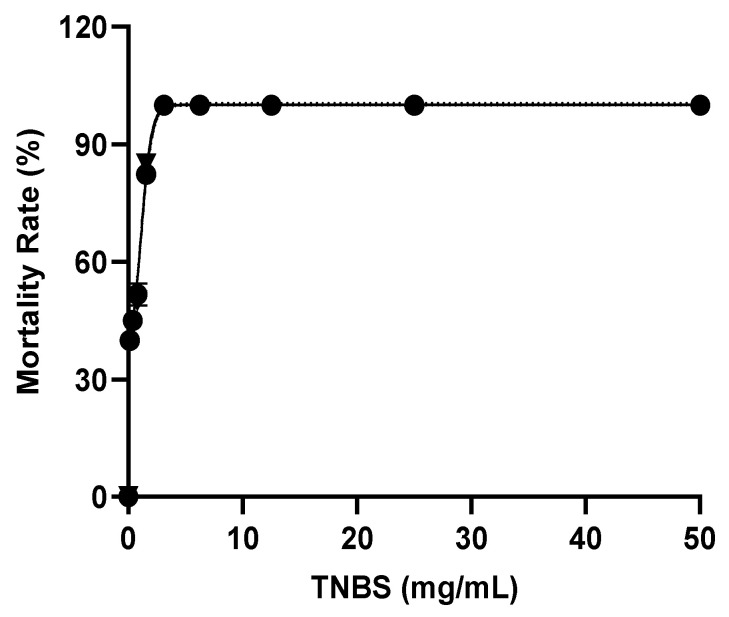
Effect of TNBS at concentrations of 0.2–50 mg/mL on zebrafish larvae (12 dpf) mortality rate. The LC_50_ value of TNBS was 0.60 mg/mL.

**Figure 7 molecules-27-08304-f007:**
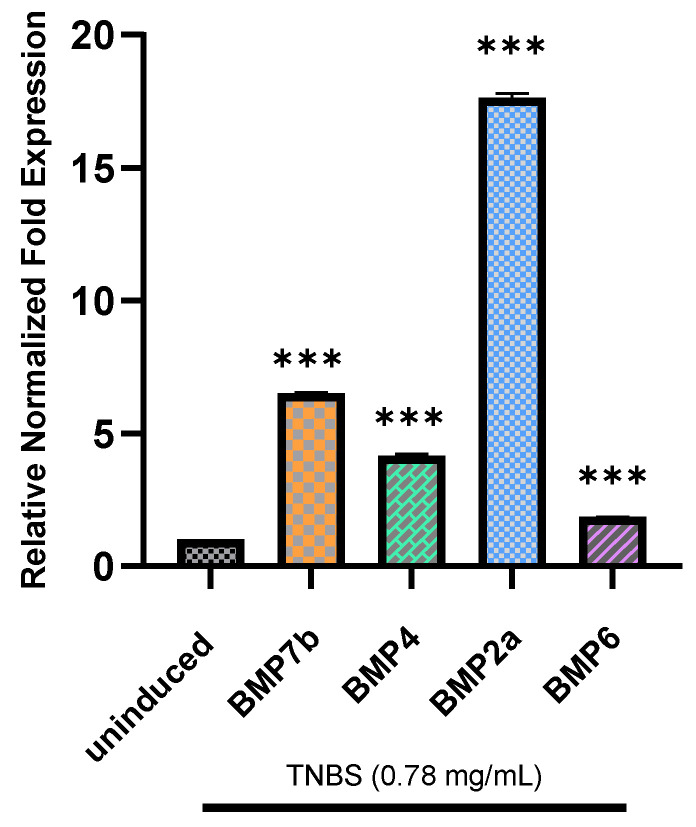
Relative normalized expression of *BMP7b*, *BMP4*, *BMP2a*, and *BMP6* induced with 0.78 mg/mL of TNBS. Thirty 12-day old larvae were pooled from each of the uninduced and TNBS-induced groups for total RNA extraction. All genes were normalized against two reference genes: *GAPDH* and *β actin*. Data are expressed as the mean ± S.E.M. of three replicates. Values of *** *p* < 0.001 were considered significantly different from the uninduced group.

**Figure 8 molecules-27-08304-f008:**
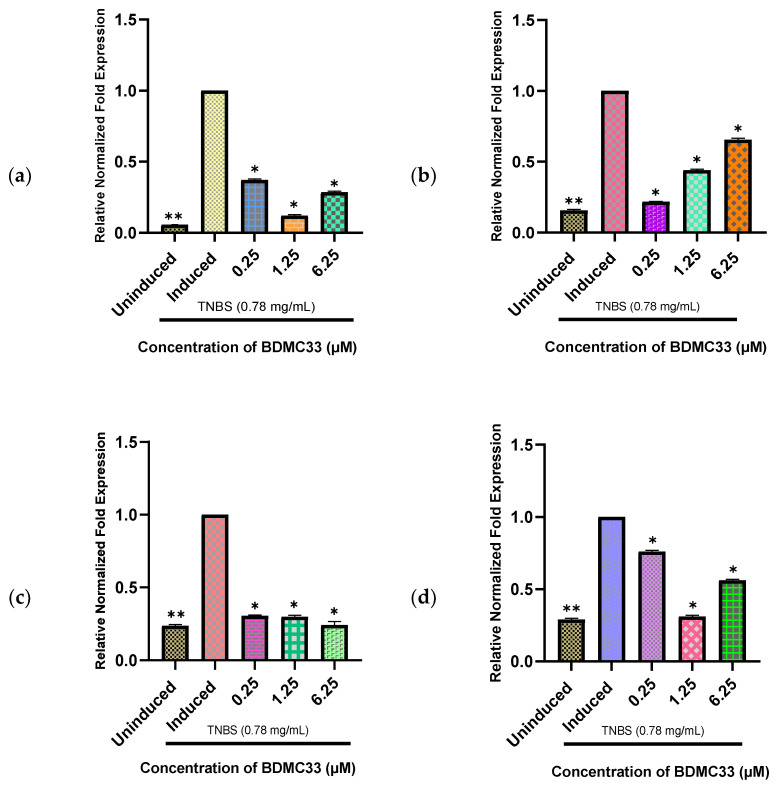
Relative normalized fold expression in (**a**) *BMP2a*, (**b**) *BMP7b,* (**c**) *BMP4*, and (**d**) *BMP6* genes induced with 0.78 mg/mL of TNBS and treated with BDMC33 at three different concentrations (0.25, 1.25, and 6.25 μM). Thirty 12-day-old larvae were pooled from each of the uninduced, TNBS-induced, and TNBS-induced with treatment (BDMC33) groups for total RNA extraction. All genes were normalized against two reference genes: *GAPDH* and *β actin*. Data are expressed as the mean ± S.E.M. of three replicates. Values of * *p* < 0.05 and ** *p* < 0.01 were considered significantly different from the induced group.

**Table 1 molecules-27-08304-t001:** Physicochemical properties of the 38 BMP family members identified from the zebrafish genome.

Physicochemical Properties	Values
Molecular weights	40,201.62–54,217.72 Da
Isoelectric point (pI)	6.13–9.85
Total number of negatively charged values Asp + G/Glu	1883
Total number of positively charged residues Arg + Lys	1605
Protein Length	347–501 AA
Instability Index	41.07–81.74
Aliphatic Index	71.07–90.28
GRAVY	−0.225–(−0.786)

**Table 2 molecules-27-08304-t002:** List of primers used in RT-qPCR.

Accession No	Gene Name	Forward Primer	Reverse Primer	Tm
NM_131359.1	BMP2a	5′-TTCGCTAAACAACGCAAGTG-3′	5′-GAACAAGCCTGGTGTCCAAT-3′	58.3 °C
NM_131342.2	BMP4	5′-TGAGGCACAACACCTCCAAA-3′	5′-ACTTTTGCCGTCATGTCCGA-3′	61 °C
NM_001013339.1	BMP6	5′ACCTGTTTCTGCTGGAATCTC3′	5′CCCTGCATCCTTTGGACTG3′	56.9 °C
NM_131321.2	BMP7b	5′-TTGACCTTTCTCGCATCCCG-3′	5′-GCAGGTACAGCTCCCTTCTG-3′	61 °C
NM_001115114.1	GAPDH	5′-CATCTTTGACGCTGGTGCTG-3′	5′-TGGGAGAATGGTCGCGTATC-3′	59.3 °C
NM_131031.2	β Actin	5′-TACCACTTTGCCCTCCTCAC-3′	5′-GACACCCTGGCTTACATTTTCA-3′	58.6 °C

## Data Availability

Not applicable.

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
