# Peer review of "In Silico Study and Effects of BDMC33 on TNBS-Induced BMP Gene Expressions in Zebrafish Gut Inflammation-Associated Arthritis"

_molecules, 2022, doi:10.3390/molecules27238304_

Round 1
Reviewer 1 Report
Abstract: The background section of the abstract is too long. Please re-write it Introduction: It is so long and confusing. Also, the references are old. Why authors did not use the references for 2021 and 2022? Also clear what is the gap. what was your solution for this gap?
Results: Good
Methods: Good
Discussion: The discussion is confusing and also the references are old. Please use more recent studies to compare with your results.
Conclusion: I think the conclusion needs some modification. You should conclude your own results and not more. In the final paragraph of the conclusion, you may suggest some future perspectives.
The similarity of text: The similarity in some sections (especially the introduction and methods) is high. Please decrease it. I attach the similarity check file.

Author Response
Abstract: The background section of the abstract is too long. Please re-write it Introduction: It is so long and confusing. Also, the references are old. Why did authors not use the references for 2021 and 2022? Also clear what is the gap. what was your solution for this gap?
- The background section of the abstract and introduction was revised. References have also been updated and cited most recent papers; however, some references are difficult to change as those papers are key evidence and perfectly align with our works for example, paper of Flemming et.al, 2010, Maric et. al, 2011 and Lee et. al, 2009.
Results: Good
Methods: Good
Discussion: The discussion is confusing and also the references are old. Please use more recent studies to compare with your results.
- The discussion section has been updated and in case of references same goes like above mentioned explanation.
Conclusion: I think the conclusion needs some modification. You should conclude your own results and not more. In the final paragraph of the conclusion, you may suggest some future perspectives.
- Conclusion has been revised according to suggestions and added future perspectives.
The similarity of text: The similarity in some sections (especially the introduction and methods) is high. Please decrease it. I attach the similarity check file.
- Similarity of text has also been reduced.
Author Response
- The aims and hypothesis of the study were revised according to suggestions. In our study, we’ve used BDMC33 which is a curcumin analogue not a curcumin supplements. This curcumin analogue has synthesized by our own research group by modifying the structure of curcumin and has shown promising anti-inflammatory properties in previous studies conducted from our lab.
Some of the concerns that need to be addressed are:
Point 1. In the Introduction section, the aim of the study should be clearly provided in the form of a paragraph, while still using numberings like(1), (2)…
- The introduction section has been revised.
Point 2. It is important to provide some of the basic physico-chemical characteristics of BDMC3 and TNBS
- The basic physico-chemical characteristics of BDMC33 and TNBS have been added that parallel to our study.
Point 3. In the section 2.5 the authors stated that “TNBS is known to induce inflammation in the gut by increasing the secretion of pro-inflammatory cytokines such as TNF-α” but no analysis of pro-inflammatory molecules was reported. To demonstrate the pro-inflammatory effect of the proposed treatment, data should be expanded to analyze the expression of TNF-a and other cytokines at gene and/or protein level.
- After discussion with all authors of this paper, we agreed to omit this information as it is not directly related to our findings.
Point 4. To better evaluate the effect of BDMC33 on TNBS-induced BMP signalling pathway the gene expression should be confirmed at protein level by means of WB or ELISA analysis.
- As it is a preliminary study of effect of BDMC33 on TNBS-induced BMP signalling pathway, we primarily focused on gene expression study. We’ll expand our investigation in the upcoming study to confirm our findings by ELISA.
Point 5. The “Methods” section should be completed. In particular:
- In the 4.4 section, please explain the survival assay in more detail.
- Section 4.4 has been updated with essential explanation.
- In the 4.7 section, please specify more, how the normalization was performed; moreover, the housekeeping genes used to normalize the results should also be reported in this section.
- Section 4.7 has also been revised and included name of housekeeping genes and explained normalization process as per required.
Point 6. Finally, in the discussion section the authors could better discuss and summarize the results obtained, referring also, to the most recent literature.
- -The discussion section has revised according to the suggestions; however, some references are difficult to change as those papers are key evidence and perfectly align with our works for example, paper of Flemming et.al, 2010, Maric et. al, 2011 and Lee et. al, 2009. We also cited some of the data, particularly explaining curcumin analogues mode of action and also determining the relationship among BMP signalling pathway with MAPK and NF-ĸβ pathway which were previously generated from our lab.
Reviewer 3 Report
This article “BDMC33 suppress TNBS-induced gut inflammation associated arthritis targeting BMP family members in Zebrafish larvae” is interesting, but suffers from the lack of key evidence and a shortage of in depth understanding and investigation. The major concer is that arthritis functions were not investigated in this study. Functional studies are crucial for the confirmation of the successful induction of the disease models and the effectiveness of the curcumin analogue. The authors did not give any clear link between arthritis and gut inflammation.
Author Response
Our study is a preliminary assessment of gut inflammation associated arthritis and further investigations will be needed for in depth understanding. In the introduction section and methodology section, we’ve mention we used TNBS to induce both gut and joint inflammation. In this current study, we don’t aim to do any functional studies rather our plan is just to induce inflammation and study the gene expression of selected BMPs and evaluate the effect of BDMC33 on this inflammatory state. Interestingly, a clear link between arthritis and gut inflammation has not been settled yet and in our study, we’ve put an attempt to explore this connection. Overall, to make our study more comprehensive, we agreed to change the title of the paper. Our newly proposed title is “In silico Study and Effects of BDMC33 on TNBS-induced BMP Gene Expressions in Zebrafish Gut Inflammation Associated Arthritis”.
Round 2
Reviewer 1 Report
Thanks for your trust. The modification based on my first report has been done by the authors. Then, I think the manuscript can be accepted.
Author Response
Thank you for accepting our manuscript.
Reviewer 2 Report
A numbers of points and concerns were addressed adequately in this second version and the present version is acceptable for publication
Author Response
Thank you for your comments and acceptance on our manuscript.
Reviewer 3 Report
I thank the authors for responding to my concerns. Unfortunately, the key evidence and functional phenotype studies (which are not that difficult and are necessary) mentioned in the 1st review is still not added.
Author Response
Padovani, B.N. et al., (2022) has successfully showed that zebrafish induced with TNBS could release TNF-α and IL-1β through NF-κB gene expression.
On the other hand, our previous study have shown that the curcumin derivative, BDMC33 inhibits the secretion of major pro-inflammatory mediators in stimulated macrophages, and includes NO, TNF-α and IL-1β through interference in both nuclear factor kappaB (NF-κB) and mitogen activator protein kinase (MAPK) signaling cascade in IFN-γ/LPS-stimulated macrophages (Ka-Heng, et al., (2012).
Since our main focus of this manuscript is to focus on the gene expression of BMP due to TNBS-induced in zebrafish, thus we don’t evaluate the pro-inflammatory cytokines and transcription factors in this study.
References
Ka-Heng, L., et al., (2012). Int J Mol Sci., 13(3), 2985-3008.
Padovani, B.N. et al., (2022). Curr Res Immunol., 3, 13-22.